# Ether Derivatives of Naringenin and Their Oximes as Factors Modulating Bacterial Adhesion

**DOI:** 10.3390/antibiotics12061076

**Published:** 2023-06-19

**Authors:** Anna Duda-Madej, Joanna Kozłowska, Dagmara Baczyńska, Paweł Krzyżek

**Affiliations:** 1Department of Microbiology, Faculty of Medicine, Wroclaw Medical University, Chałubińskiego 4, 50-368 Wrocław, Poland; 2Department of Food Chemistry and Biocatalysis, Faculty of Biotechnology and Food Science, Wrocław University of Environmental and Life Sciences, C.K. Norwida 25, 50-375 Wrocław, Poland; joanna.kozlowska@upwr.edu.pl; 3Department of Molecular and Cellular Biology, Faculty of Pharmacy, Wroclaw Medical University, Borowska 211A, 50-556 Wrocław, Poland; dagmara.baczynska@umw.edu.pl

**Keywords:** adhesion, biofilm, Bioflux, microfluidic conditions, naringenin derivatives, intestinal microflora

## Abstract

Because of the close connection between adhesion and many vital cellular functions, the search for new compounds modulating the adhesion of bacteria belonging to the intestinal microbiota is a great challenge and a clinical need. Based on our previous studies, we discovered that *O*-lkyl naringenin derivatives and their oximes exhibit antimicrobial activity against antibiotic-resistant pathogens. The current study was aimed at determining the modulatory effect of these compounds on the adhesion of selected representatives of the intestinal microbiota: *Escherichia coli*, a commensal representative of the intestinal microbiota, and *Enterococcus faecalis*, a bacterium that naturally colonizes the intestines but has disease-promoting potential. To better reflect the variety of real-life scenarios, we performed these studies using two different intestinal cell lines: the physiologically functioning (“healthy”) 3T3-L1 cell line and the disease-mimicking, cancerous HT-29 line. The study was performed in vitro under static and microfluidic conditions generated by the Bioflux system. We detected the modulatory effect of the tested *O*-alkyl naringenin derivatives on bacterial adhesion, which was dependent on the cell line studied and was more significant for *E. coli* than for *E. faecalis*. In addition, it was noticed that this activity was affected by the concentration of the tested compound and its structure (length of the carbon chain). In summary, *O*-alkyl naringenin derivatives and their oximes possess a promising modulatory effect on the adhesion of selected representatives of the intestinal microbiota.

## 1. Introduction

Cell adhesion is a phenomenon of paramount importance in various physiological processes. It affects a plethora of physiological functions of cells, e.g., growth, differentiation, and migration. Interactions during adhesion can occur either on the cell-to-cell or cell-to-extracellular matrix (ECM; extracellular matrix) pathway, which for eukaryotic cells is essential for the development of tissues, organs, and even whole organisms [1]. Non-specific factors responsible for cell adhesion include van der Waals forces, hydrogen and ionic bonds, and the presence of hydrophobic groups [2]. Cell–cell inter-adhesion is additionally stabilized by specific mechanisms, including the production of cell membrane glycoproteins, known as cell adhesion molecules (CAMs). The role of CAMs has been confirmed in many processes, both physiological and pathological. Under natural conditions, they are involved in, e.g., embryogenesis, wound healing, and the maintenance of normal tissue organization [3,4]. The negative effects of CAMs include their involvement in tumorigenesis by allowing: (1) progression (growth of the primary tumor by affecting cell differentiation); (2) tissue invasion (morphological abnormalities and disorganization of the cytoskeleton); and (3) metastasis (involvement in detachment of tumor cells from the neoplastic mass and migration to other organs) [5,6,7].

Adhesion is, however, not limited to eukaryotic cells. This phenomenon is also of high importance for prokaryotic cells, which, using a number of factors, e.g., motile cilia, fimbriae, lipopolysaccharide, and outer membrane proteins [8], easily reach the epithelial surface. A model example of the relevance of the ligation process of microbes with cell receptors is the gastrointestinal tract, the largest organ of our body. There are many factors preventing the adhesion of microbes to the gastrointestinal tract, including mucus production, intestinal peristalsis, antibodies, and other components of the non-specific immune response. This greatly underscores the importance of this environment in the context of modulating microbial adhesion. However, despite these processes, microorganisms constituting the gastrointestinal microbiome can easily cope with this microenvironment and overcome the above-mentioned disadvantages, reaching a final biomass of 10^13^–10^14^ cells [9]. Over the years, it has become clear that the gastrointestinal microbiota coordinates the proper functioning of metabolism, immunity, and the development of the human body. Intestinal bacteria exhibit a number of positive effects on the health of the host, such as the synthesis of vitamins (vitamin K or B group vitamins), degradation and detoxification of toxic and mutagenic compounds, maintenance of intestinal epithelial integrity (production of short-chain fatty acids), adsorption of electrolytes and mineral salts (sodium, calcium, magnesium, potassium), as well as production of compounds with bactericidal activity (e.g., bacteriocins) [10,11]. However, in this unique environment, the adhesion process of potentially pathogenic microorganisms can also negatively influence the human body. It initiates the production of pro-inflammatory cytokines, which stimulate the development of an inflammatory, destructive cascade in the intestinal mucosa. Simultaneously, effective adherence of microbes increases the risk of their transition into the deeper layers of the mucosa and the production of systemic infections. In that respect, of particular interest are reports highlighting the involvement of intestinal bacteria in the development of some severe neurodegenerative conditions, such as Parkinson’s and Alzheimer’s disease [12,13]. Because of the double-edged nature of the adhesion process (involvement in cancer progression and infectious diseases or maintaining the physiology of the host), proper control of this process seems to be of high scientific relevance. So far, several adhesion-modulatory strategies have been described, including: (1) modification of the microbial surface [14], (2) blockage of adhesin biosynthesis [15], (3) interference with adhesin modification processes [16], or (4) the use of methods blocking interactions between adhesins and cell receptors, e.g., anti-adhesion antibodies [17]. It is worth highlighting that adhesive molecules also have a positive effect on the immune response. This has allowed the development of anti-adhesion strategies aimed at modulating disorders of the immune system, including asthma, psoriasis, Crohn’s disease, multiple sclerosis, inflammatory bowel disease (IBD), and cancer [18,19,20,21,22,23].

In the last decade, it has become very popular in the scientific community to screen various substances for their ability to modulate the virulence of different pathogens, including interference with the production of adhesins. There are already the first indications that compounds from natural sources (e.g., lycopene) affect the adhesion process both in vitro and in vivo [24]. Therefore, it is advisable to find new substances that have a modulating effect on the adhesion process. A very promising group of compounds commonly found in nature are flavonoids. They exhibit a wide range of positive effects on human health. They modulate the activity of the immune system and present anti-cancer, anti-inflammatory, anti-atherosclerotic, and neuroprotective effects [25]. Previous studies performed by our team underscore the attractiveness of bioflavonoid derivatives. Indeed, we demonstrated the antimicrobial activity of *O*-alkyl derivatives of naringenin and their oximes against multidrug-resistant strains, e.g., *Staphylococcus aureus* MRSA (methicillin-resistant *S. aureus*), *Enterococcus faecalis* VRE (vancomycin-resistant *Enterococcus*), and clarithromycin-resistant *Helicobacter pylori* [26]. In addition to this, our latest report showed the anticancer activity of these compounds against the human colorectal adenocarcinoma cell line (HT-29) [27].

Accordingly, the aim of the present study was to test the ability of *O*-alkyl derivatives of naringenin and their oximes to modulate the adhesion of selected microorganisms colonizing the gastrointestinal tract to cell lines derived from this organ.

## 2. Results

In our investigations, we determined the impact of *O*-alkyl derivatives of naringenin (**1a**–**10a**) and their oximes (**1b**–**10b**) on the adhesion of two representative strains of gut microbiota to HT-29 and 3T3-L1 cell lines. All compounds that were used in our research are presented in Table 1. The chemical data and methods of obtaining these derivatives were described by us in previous works [26,27].

In the first stage of determining the activity of our newly synthesized compounds, we decided to check the effect of their different concentrations on the adhesive properties of two selected bacterial species: *E. coli*, a commensal representative of the intestinal microbiota, and *E. faecalis*, a bacterium that naturally colonizes the intestines but has disease-promoting potential. To better reflect the variety of real-life scenarios, we performed these studies using two different intestinal cell lines, i.e., the physiologically functioning (“healthy”) 3T3-L1 cell line and the disease-mimicking, cancerous HT-29 line.

When *E. coli* was co-incubated with the 3T3-L1 line, we noticed a very interesting correlation between the structure/chain length of tested compounds and their ability to modulate a physical bacteria-cell line interaction (Figure 1). We—observed that short—chain **a** compounds very strongly promoted the adhesion of *E. coli* to the 3T3-L1 line (e.g., **1a** at 1–50 µg/mL increased attachment by 3 to 16 times). The longer the chain of these compounds, the weaker the effect was, until the moment when bacterial adhesion was disturbed by substances **9a** (≈1.1- to 2-fold for 10–100 µg/mL) and **10a** (≈2-fold, regardless of the concentration used). Interestingly, we observed a completely opposite effect for **b** compounds. In this case, long-chain compounds promoted adhesion (**10b** was the strongest and induced this process 1.5- to 5.5-fold), while short-chain ones significantly reduced this phenomenon (e.g., **1b** at 1 µg/mL decreased attachment to 63%, while at a concentration of 25 µg/mL adherence was practically not observed). For all the tested compounds promoting adhesion, both from the **a** and **b** groups, we noticed attachment-inducing activity only when using low-level concentrations.

For *E. coli* co-incubated with the HT-29 line, the obtained results were much more homogeneous—all the tested substances reduced the bacterial adhesion to eukaryotic cells (Figure 1). In this case, the exceptions were substances **1a**, **3b**, **4b**, **5b**, **6b**, and **10b** at 50 µg/mL as well as **4b** and **5b** at 25 µg/mL. The most interesting results in this regard were obtained for compound **4b**, because at concentrations within the range of 1–10µg/mL or 75–100 µg/mL the level of adherence was very low (often only a few percent), while at concentrations equal to 25 µg/mL and 50 µg/mL adhesion was 17- and 5-fold higher than in the control.

The results described in the above paragraphs suggest that the effect of tested compounds on *E. coli* is: (a) dependent on the tested cell line and often has a positive effect on adhesion to the physiological 3T3-L1 line; (b) in general terms, in both tested cell lines, the positive effect of low concentrations and the negative effect of high concentrations on the adherence of *E. coli* were noticed, while co-incubation studies of this bacterium with the HT-29 line indicate also that precisely selected, high concentrations of some compounds can strongly induce bacterial adhesion to the cell line.

In an alternative research scenario, taking into account the co-incubation of *E. faecalis* with eukaryotic cells and the presence of various concentrations of tested compounds, the effect of the cell line was not as significant as in the case of *E. coli*. When we used the 3T3-L1 line, we again noticed the relationship between the length of the side chain of compounds and their impact on bacterial adhesion (Figure 1). This effect was, however, visible exclusively when increasing the side chain only from the C-7 position of naringenin (**1a**/**1b** vs. **3a**/**3b** vs. **5a**/**5b** vs. **7a**/**7b** vs. **9a**/**9b**), but not from both C-7 and C-4′ sites of naringenin simultaneously (**2a**/**2b** vs. **4a/4b** vs. **6a**/**6b** vs. **8a**/**8b** vs. **10a**/**10b**). For both tested groups of compounds (annotated as **a** and **b**), we observed that the extension of the side chain has a negative effect on the pro-adhesive activity of these substances. For example, the compounds **1a** and **3a** as well as **1b** and **3b** had a much more beneficial effect on the adhesion of *E. faecalis* (at 1–10 µg/mL they increased this process 1.5- to 3-fold) than **7a** and **9a** as well as **7b** and **9b**, which often actually limited the attachment.

When using the HT-29 line, co-incubation with *E. faecalis* showed a similar relationship between the structure of the tested compounds and their effect on adhesion as presented previously (Figure 1). We noticed that substances with the attachment of the eight- or nine-carbon chain(s) in the C-7 and C-4′ positions of naringenin (**6a**–**8a** and **6b**–**8b**) promoted the adhesion of bacteria to eukaryotic cells (depending on the concentration used, up to two times), while both compounds with shorter or longer side chains reduced this process.

In general, the results described above indicate that the effect of tested substances on *E. faecalis* is: (a) dependent on the tested cell line, although this effect is much less significant than in the case of *E. coli*; (b) influenced by the compounds’ structure; (c) in simple terms, a positive effect of low concentrations and a negative effect of high concentrations were observed; similarly to *E. coli*, when co-incubating *E. faecalis* with the HT-29 line, some precisely selected, high concentrations of the tested compounds can strongly increase the adhesion of this bacterium to the cell line.

In the next step, we decided to determine whether the tested naringenin derivatives negatively affect the viability of the examined intestinal cell lines. The cytotoxic effect of *O*-alkyl naringenin derivatives (**1a**–**10a**) and their oximes (**1b**–**10b**) against the human colon cancer line (HT-29) and “healthy” murine fibroblasts (3T3-L1) is reported as survival index (SI) IC_50_ [µg/mL], being a concentration able to induce 50% inhibition of cell proliferation. The obtained data are presented in Table 2 and were compared with the values determined for the reference substances—naringenin (**NG**) and naringenin oxime (**NGOX**).

Naringenin and its oxime showed no proliferative effects against both cell lines, 3T3-L1 and HT-29, over the concentration range tested. The IC_50_ value for these two model compounds was >100 µg/mL.

Among the tested naringenin derivatives lacking an oxime group, cytotoxic activity against the 3T3-L1 line was demonstrated for monosubstituted *O*-alkyl derivatives of naringenin, e.g., **1a**, **3a**, **5a**, **7a**, and **9a**. The same compounds, with the exception of **9a**, also showed antiproliferative activity against the HT-29 line. It is interesting to note that the cytotoxic effect was observed only for compounds having one carbon chain attached at the C-7 position of the naringenin ring. Furthermore, a correlation between the number of carbon atoms and the level of cytotoxic activity was observed. Indeed, the shorter the attached carbon chain, the greater the effect of the compound on the proliferation of the adenocarcinoma cells examined. In fact, the IC_50_ value for compound **1a**, which contains a hexyl group attached to the C-7 position of naringenin, was 25.39 µg/mL, which was 1.8 times lower than compound **7a** (IC_50_ = 46.72 µg/mL), which possesses a nonyl moiety.

In contrast, against the 3T3-L1 line, compounds having chains with 6 to 9 carbon atoms showed an effect on proliferation at a level 3–4 times higher than compounds with longer chains (e.g., **9a** having 10 carbon atoms). Naringenin derivatives with two alkyl chains attached (the second at the C-4’ position) with different numbers of carbon atoms (compounds **2a**, **4a**, **6a**, **8a**, and **10a**) showed no effect on the proliferation of the tested cell lines in the tested concentration range.

In the case of naringenin derivatives bearing an additional oxime group (compounds **1b**–**10b**), an effect on 3T3-L1 cell proliferation was also noted only for monosubstituted compounds, e.g., **1b**, **3b**, **5b**, and **7b**. For them, IC_50_ values were in the range of 32.12 µg/mL to 49.93 µg/mL. A similar relationship was observed for the human adenocarcinoma cell line because the same compounds (**1b**, **3b**, **5b**, and **7b**) showed cytotoxic effects against this cell line. The IC_50_ value depended on the length of the attached chain and was higher for the oxime derivatives with a longer alkyl chain attached to the naringenin ring. On the other hand, both **4b** and **10b**, oximes of *O*-alkyl naringenin derivatives with attached alkyl chains at the C-7 and C-4’ positions, affected cell differentiation of the HT-29 line with comparable IC_50_ values of 43.35 and 48.23 µg/mL, respectively. In contrast, the other di-substituted compounds, e.g., **2b**, **6b**, and **8b**, showed no harmful effects against human colon cancer cells in the concentration range tested (IC_50_ > 100 µg/mL).

Based on the results of the experiments presented above (adhesion in static conditions and cytotoxicity against cell lines) as well as our detailed review of the literature indicating the highest biological activity of flavonoids with a C8–C10 side chain [28,29,30,31,32,33], at the final proof-of-concept stage of this research, compound **8b** at 50 µg/mL was administered. In this context, our priority was the ability of **8b** to strongly enhance the adhesion of *E. coli* to the physiological 3T3-L1 cell line without causing any cytotoxic effects. The choice of concentration was, in turn, related to our willingness to obtain a noticeable biological effect of this substance in an experimental model with high dynamics of physicochemical conditions—microfluidic studies.

Applying microfluidic conditions, we observed that the use of **8b** at 50 µg/mL had a very strong, pro-adhesive effect on *E. coli* with respect to both cell lines tested. Here, for the physiological 3T3-L1 cell line, the ratio of the area of bacterial biomass (BB) adhered to the surface of the eukaryotic cells (EC) [the BB/EC ratio] was 0.6 ± 0.12 vs. 0.37 ± 0.13 for compound-exposed and control samples, respectively (Figure 2). For the tumor-altered HT-29 cell line, the BB/EC ratio was 1.03 ± 0.06 vs. 0.65 ± 0.07 for substance-exposed and control samples, respectively (Figure 2). Interestingly, under identical environmental conditions, we did not detect such an effect for *E. faecalis*. In that case, for both cell lines, the effect was neutral (the BB/EC ratio for treated and control samples was: 0.28 ± 0.15 vs. 0.29 ± 0.11 for 3T3-L1 and 1.36 ± 0.08 vs. 1.55 ± 0.15 for the HT-29 cell line) (Figure 2). Representative photographs showing the adhesion of both tested bacteria to the surface of the physiological cell line, 3T3-L1, with and without exposure to **8b**, are shown in Figure 3. The results obtained by us in the current set of microfluidic experiments indicate a high selectivity in the pro-adhesive effect of the chosen substance. As demonstrated in the example above, this activity can promote the adherence of commensal microorganisms with a beneficial effect on the host’s gut (e.g., physiological strains of *E. coli*) without a negative, dysbiotic effect on other representatives of the microbiota (including opportunistic bacteria, e.g., *E. faecalis*).

## 3. Discussion

Naringenin is an organic flavonoid compound that is widely distributed in nature. The largest source of naringenin is citrus fruits (grapefruit, orange, and tangerine), although it is also present in smaller amounts in grapes, cherries, fenugreek, and Greek oregano, as well as in coffee, tea, and red wine. The literature review confirms the broad-spectrum, health-promoting effect of this flavonoid on the human body. It has been shown that such activity is closely related to the affinity of naringenin for scavenging reactive oxygen species and increasing the antioxidant defense of the host. This strictly translates into its anti-inflammatory effect and determines its anti-atherosclerotic and neuroprotective features. Additionally, it has been documented that naringenin also has anti-microbial and anti-cancer activity [25,28].

The cell membrane is one of the main target sites of flavonoids against microorganisms [34]. As observed in mechanistic studies on flavonoids, these compounds localize themselves within the hydrophobic fractions of the lipid membrane bilayer and lead to change in its fluidity and stiffness [35,36]. For this reason, the chemical structure of flavonoids and their ability to modulate membrane fluidity are important parameters affecting their biological activity [36]. For naringin and naringenin, two flavonoids relevant to the current study, the capacity to preferentially localize in the polar lipid membrane bilayer and exert an ordering effect on the hydrophobic region of this compartment has been demonstrated [37,38]. Such activity of flavonoids, including naringenin, contributes to the loss of the fluidity of cell membranes [35]. It is worth mentioning, however, that this membrane-solidifying effect of naringenin is revealed only at higher concentrations of ≥2.5 µg/mL [39]. Quite obviously, such changes in cell membrane fluidity affect the physiological properties of microorganisms [40]. In the study of Cazzola et al. [41], a close correlation was observed between the increase in *E. coli* adhesion to eukaryotic cells and the decrease in cell membrane fluidity of these bacteria. In another study carried out on four selected bacterial species, it was shown that, together with the entry into the biofilm (sedentary) phase, microorganisms accumulate saturated fatty acids in their cell membranes, leading to a decrease in their membranes’ fluidity [42]. Identical observations were made in another article [43], where, using *Pseudomonas aeruginosa* as a model organism, the participation of extracellular vesicles in membrane stiffening and promotion of biofilm formation was additionally noticed. The above-described observations seem to perfectly complement our team’s observations made in this paper, as we noticed that exposure of bacteria to naringenin and the panel of its derivatives is accompanied by modifications in the ability of the tested bacteria to adhere to host cells. Undoubtedly, this effect was dependent on many factors, including both the chemical structure of the compound and its concentration, as well as the type of bacteria or intestinal cell line used. Nevertheless, under conditions of dynamic medium flow, we noticed the ability of the selected naringenin derivative **8b** to promote the adhesion of *E. coli* to both cell lines while having no impact on *E. faecalis*. Based on our knowledge of the cell structure of both species of bacteria (Gram-negative *E. coli* and Gram-positive *E. faecalis*), we suspect that the physical proximity between the cell membrane of *E. coli* and the culture medium with the tested compound had a direct impact on the results obtained. Gram-positive bacteria have no outer membrane, and the only one that exists is physically separated from the external environment by a densely cross-linked layer of peptidoglycan [44,45]. The physiology of Gram-negative bacteria is more closely related to the presence of cell membranes as they produce two of them (inner and outer membranes) [46,47], and therefore we postulate that the action of naringenin derivatives may have a greater impact on changes in *E. coli* membrane fluidity and its adhesive properties.

When assessing the influence of naringenin derivatives on the adhesive properties of bacteria, one cannot ignore the role of their chemical structure in modulating the physiology of microorganisms. In this context, consideration of our results obtained for the panel of chemical modifiers of naringenin (alkoxy derivatives and their oximes) seems very helpful. In general terms, we noticed that the appearance of the oxime residue affects the activity of the tested compounds, but this effect was variable and largely dependent on chemical modifications within the structure of naringenin, especially on the length of the alkyl chain attached. In our opinion, much more interesting observations were obtained with respect to the correlation between the presence of aliphatic chains in the C-7 and C-4′ positions and the pro-adhesive properties of these substances. Here, we noticed that the highest biological activity was most often found for naringenin derivatives with 8- to 10-carbon side chains. Both shorter- and longer-chain compounds had reduced activity. When considering the causes of this phenomenon, it is worth focusing on the physicochemical implications of the above chemical modifications. Lipophilicity is one of the key parameters determining the activity of flavonoids [34,48]. Therefore, hydrophobic substituents (including alkyl chains) usually increase the biological activity of substances [48]. The elongation of the side chain increases the hydrophobicity of the compound and, consequently, leads to a stronger affinity for biological membranes [34,48]. However, it is suggested that excessive elongation of compounds’ side chains is related to the decline of their optimal solubility, eventually leading to a drastic decrease in their biological activity (the so-called “cut-off effect”) [29,34]. These observations are in great agreement with the results obtained by other research teams, suggesting that the presence of medium-chain substituents has the most beneficial effect on the increase in the flavonoids’ activity [29,30,31,32,33]. To sum up, in the current article, we proved that the chemical modification of naringenin and attachment of medium-length side chains have a positive effect on their pro-adhesive properties towards bacteria.

The gastrointestinal tract is the most heterogeneous and microbiologically dense organ of the human body. A long residence time in the intestines is a characteristic pharmacokinetic feature of flavonoids. For this obvious reason, it has been proven that during this prolonged period, they interact with a variety of representatives of the microbiota [49]. This type of interaction translates into two important aspects. Firstly, microorganisms are involved in the transformation of naringin (inactive form) into a host-beneficial naringenin (active form) [50]. Thus, the gastrointestinal microbiota plays a critical role in the bioavailability of this bioflavonoid in the human body. On the other hand, the presence of flavonoids, including naringenin, is not indifferent to the microbiota residing in the intestines. Many flavonoids (e.g., naringin, naringenin, hesperetin-7-*O*-glucoside, prunin, isoquercitrin, hesperidin, rutin, and quercetin) have been shown to interact with the vast majority of microorganisms belonging to the intestinal microbiota. In this way, flavonoids present a modulatory effect on both the structure and qualitative composition of the gastrointestinal microbiota [51]. The interaction of naringenin with representatives of the natural intestinal flora and their metabolites has been shown to improve animal health, e.g., in polycystic ovary syndrome (PCOS) [52] and non-alcoholic fatty liver disease (NAFLD) [53]. Next-generation sequencing showed that administration of naringenin increases the abundance of a plethora of health-promoting bacterial species that are part of the gastrointestinal microbiome, including *Lactobacillus* spp., *Faecalibacterium* spp., *Butyricicoccus* spp., *Coprococcus* spp., and *Roseburia* spp. [54]. It is also worth emphasizing that the “metagenome”, a whole pool of genes of the microbiome, encodes information about the metabolism of different classes of carbohydrates, amino acids, or xenobiotics, sometimes being the only source for their degradation and absorption in a human organism [55,56]. Therefore, it is significant that naringenin, by enriching the microbiome of the gastrointestinal tract with species of “beneficial” bacteria, promotes homeostasis of the host not only by regulating inflammatory reactions but also by increasing its digestive abilities. In addition, through this unique function, it prevents intestinal dysbiosis and thus protects against the development of inflammatory bowel disease (IBD) (Crohn’s disease, ulcerative colitis, and indeterminate colitis), irritable bowel syndrome (IBS), and celiac disease.

## 4. Materials and Methods

### 4.1. Naringenin Derivatives

Naringenin used for the synthesis of ether derivatives (**1a**–**10a**) and their oximes (**1b**–**10b**) was purchased from Sigma-Aldrich Co. (St. Louis, MO, USA). The *O*-alkyl derivatives of naringenin and oximes were synthesized according to methods described by us in previous works [26,27]. Briefly, to obtain ether derivatives **1a**–**10a**, to naringenin dissolved in organic solvent (anhydrous acetone or DMF; Chempur, Piekary Śląskie; Poland), potassium carbonate and appropriate alkyl iodide in a molar ratio of 1:1.5:5 were added, respectively. The reaction mixture was kept on a magnetic stirrer at room temperature (when the reaction was performed in DMF) or at 45 °C (when the reaction was performed in anhydrous acetone). The progress of the reaction was monitored by thin-layer chromatography (TLC), and after observing two products (mono- and di-*O*-alkyl derivatives), the crude mixture was extracted with an organic solvent (ethyl acetate or diethyl ether). The organic fractions were collected and concentrated on a vacuum evaporator, and the products were separated by liquid column chromatography.

In the next step, compounds **1a**–**10a** were modified into their oximes **1b**–**10b**. Briefly, the *O*-alkyl derivative was dissolved in anhydrous ethanol, and then hydroxylamine hydrochloride and anhydrous sodium acetate in a molar ratio of 1:3:3 were added, respectively. After complete conversion of substrate, the reaction mixture was poured into ice water, and precipitated white products were collected and purified by liquid column chromatography. Using the same method, naringenin oxime (**NGOX**) was obtained.

### 4.2. Prokaryotic Cells

The study used two reference strains, *Escherichia coli* K12 (ATCC 10798) and *Enterococcus faecalis* (ATCC 29212), obtained from the American Type Culture Collection. Both strains were stored in trypticase soy broth (TSB; OXOID, Basingstoke, UK) with the addition of 30% glycerol at −70 °C in the museum resources of the Department of Microbiology at the Wroclaw Medical University.

Strains were reactivated from deep freeze by inoculation into TSB and an overnight incubation under shaking conditions (MaxQTM6000 incubator shaker, Thermo Scientific, Waltham, MA, USA) at 125 rpm and 37 °C. The purity of the strains was assessed using enriched media: MacConkey (MC, OXOID, Basingstoke, UK) for *E. coli* and Columbia Agar (CA, Becton, Dickinson and Company, San Diego, CA, USA) for *E. faecalis*. A fresh 18–20 h culture was prepared for each experiment on tryptic soy agar (TSA, OXOID, Basingstoke, UK).

For studies determining the interaction of bacteria with eukaryotic cells, suspensions of the tested strains were prepared in Luria broth (LB, Becton, Dickinson and Company, USA) medium with a bacterial density of 6 × 10^8^ CFU/mL.

### 4.3. Cell Culture Procedure

The human adenocarcinoma cell line, HT-29 (ATCC HTB-38), was cultured in αMEM medium (IITD PAN) supplemented with 10% fetal bovine serum, FBS (Sigma-Aldrich, Taufkirchen, Germany), 100 U/mL penicillin, 100 µg/mL streptomycin antibiotic solution (Sigma-Aldrich, Taufkirchen, Germany), and 2 mM glutamine solution (Sigma-Aldrich, Taufkirchen, Germany). Cells from less than 20 passages were used for the study. In turn, preadipocytes from mouse embryos with the morphology of fibroblast cells, line 3T3-L1 (ATCC CL-173), were cultured in DMEM high-glucose medium (Life Technologies Corporation, Carlsbad, CA, USA) with the same supplements as above. Cells from less than 14 passages were used for the study. Both tested cell lines were grown at 37 °C in 5% CO_2_.

### 4.4. SRB Assay

The effect of *O*-alkyl derivatives of naringenin and their oximes on eukaryotic cells was checked using a slightly modified SRB test described earlier [57]. Firstly, HT-29 (1 × 10^5^ cells/mL) and 3T3-L1 (5 × 10^4^ cells/mL) cell lines were incubated in adherent microtiter plates under the conditions described above. The cells were then treated for 2 h with the tested compounds **1a**–**10a** and **1b**–**10b** at concentrations of 1, 5, 10, 25, 50, 75, and 100 µg/mL each. After this time, the medium was removed and replaced with one containing 50% (wt/vol) trichloroacetic acid (TCA; Sigma-Aldrich, Taufkirchen, Germany). Following 1 h of incubation at 4 °C, the plate was rinsed with water, and then an acidic solution of 0.4% sulforhodamine B (SRB) (Sigma-Aldrich, Taufkirchen, Germany), was added to the dried wells and incubated in the dark for half an hour. At the end of this time, the SRB solution was replaced by 1% (vol/vol) acetic acid, and the stained protein precipitate was dissolved in 10 mM Tris base solution (pH 10.5). Absorbance was measured at 560 nm in a plate reader (GloMax Discover Microplate Reader, Promega).

The cytotoxic effect of *O*-alkyl derivatives of naringenin and their oximes on eukaryotic cells was expressed as the concentration of the tested compounds at which 50% growth inhibition of the tested cell lines was observed. Values were presented in the form of the survival index (SI)–IC_50_. All determinations were repeated in three independent experiments, each performed in three technical replicates. Moreover, the effect of DMSO (Sigma-Aldrich, Taufkirchen, Germany), the solvent present in all the initial solutions of the tested compounds, on the growth of the cell lines used in the study was also verified.

### 4.5. Adhesion Assay

The adhesion ability of the tested bacterial strains to HT-29 and 3T3-L1 cell lines was determined by an in vitro adhesion assay as described previously [58]. Eukaryotic cells were infected with prokaryotic cells in the presence of the examined compounds in pre-selected concentrations at an MOI (multiplicity of infection) ratio of 50. Cells were exposed to the tested strains and compounds for 2 h. After this time, unbound bacteria were removed by washing with PBS (Life Technologies Corporation, Carlsbad, CA, USA). The lysates acquired by cell lysis (0.5% Triton X-100 in LB): 90 min, 37 °C, 5% CO_2_ atmosphere, were then seeded onto TSA and MC media for *E. faecalis* and *E. coli*, respectively. The assays were performed in at least three independent experiments (cells from subsequent passages), each in triplicate. A control without tested compounds was set for each strain.

### 4.6. Bioflux

Determination of the impact of a dynamic medium flow on the adhesive capacity of bacteria was performed using the Bioflux 1000Z (Fluxion, San Francisco, CA, USA) with a coupled microscope environmental chamber (Pecton Incubator XL S1, Carl Zeiss, Jena, Germany), enabling the maintenance of 37 °C and microaerophilic conditions. At the very beginning, all channels of dedicated microfluidic plates (Fluxion, San Francisco, CA, USA) were flushed with 100 µL of αMEM or DMEM high-glucose medium with supplements (further named “cell line medium”; for details of the supplementation see “Cell culture procedure”) applying an intensive flow of 10 dyne/cm^2^ for 10 s. After unblocking the lumen of the channels, the inlet and outlet wells were emptied, and 20 µL of 100 µg/mL fibronectin solution (Sigma-Aldrich, Taufkirchen, Germany) was added to each inlet channel. The medium was turned in the outlet direction at a flow rate of 1 dyne/cm^2^ for 1 min. After this step, the plate was left for a 1 h incubation period to allow fibronectin to be absorbed on the surface of the microcapillaries. After the incubation period, the microcapillaries were rinsed again with 100 µL of the cell line medium at 10 dyne/cm^2^ for 10 s to remove excess non-attached fibronectin. The outlet wells were emptied and then refilled with 100 µL of the cell line medium containing 3T3-L1 or HT-29 cells (10^8^ cells/mL). An outlet-to-inlet medium flow of 5 dyne/cm^2^ was turned on for 5 s, then the flow was stopped, and the cells were incubated for 0.5 h to allow them to effectively adhere to the surface of the microcapillaries. After this time, 1 mL of the cell line medium was added to the inlet wells, and the inlet-to-outlet flow was adjusted to 0.5 dyne/cm^2^ for 24 h, thus allowing the growth of the cell line monolayer. After one day of incubation, the inlet and outlet wells were emptied, and then 1 mL of the cell line medium [without antibiotics] containing one of the tested bacterial strains (*E. coli* or *E. faecalis*; 10^8^ CFU/mL) and a selected flavonoid, **8b** (50 µg/mL), was added to the inlet wells. In the control sample, the above compound was not added (the cell line medium [without antibiotics] together with bacteria). In regard to this, in both tested and control samples, the cell line-to-bacteria ratio was maintained at MOI = 50. After this, the medium was set to flow from inlets to outlets at 0.5 dyne/cm^2^ for 4 h. In the next step, the inlet wells were emptied, and 100 µL of saline solution containing a mix of two fluorescent dyes, SYTO9 (L10316, ThermoFisher, Waltham, MA, USA) to visualize bacterial biomass and DAPI (62248, ThermoFisher, Waltham, MA, USA), to detect eukaryotic cells, was added to each well. The inlet-to-outlet flow was turned on at 0.5 dyne/cm^2^ for 10 min, followed by a 0.5 h incubation to induce the absorption of the dyes by cells. After completing this stage, photographs of the microcapillaries containing the tested biological materials were taken using an inverted fluorescence microscope (GabH, Jena, Germany) and the Bioflux Montage software (Fluxion, San Francisco, CA, USA). When determining the area occupied by bacterial cells adhered to the surface of eukaryotic cells, the entire bacterial biomass attached was taken into account (vertically and horizontally). These calculations did not include bacterial cells adhering only to the surface of the microcapillaries. The tests were performed in three biological repetitions with three technical replications (constituting images from different fragments of the examined microcapillaries).

### 4.7. Statistical Analysis

Statistical analysis was performed using GraphPad Prism version 9 (GraphPad Co., San Diego, CA, USA). The normality of the distribution was checked by the Shapiro–Wilk test. As all values were normally distributed, Student’s *t*-test was further used. The results of statistical analyses were considered significant for values with *p* < 0.05.

## 5. Conclusions

The present study showed that the synthesized naringenin derivatives modulate the adhesion of selected intestinal microbiota to the tested cell lines. It was noticed that this activity was affected by the concentration of the tested naringenin derivatives and their structure (length of the carbon chain). To better reflect the variety of real-life scenarios, the current research on *O*-alkyl derivatives of naringenin was focused not only on different intestinal cell lines (physiological and cancerous) but also on both static and microfluidic conditions. Therefore, the promising results obtained here provide a basis for in vivo animal studies to clarify the effect of these bioflavonoids on the gut environment. This may provide valuable information on the bioavailability and effect of these compounds on the whole intestinal microbiome.

## Figures and Tables

**Figure 1 antibiotics-12-01076-f001:**
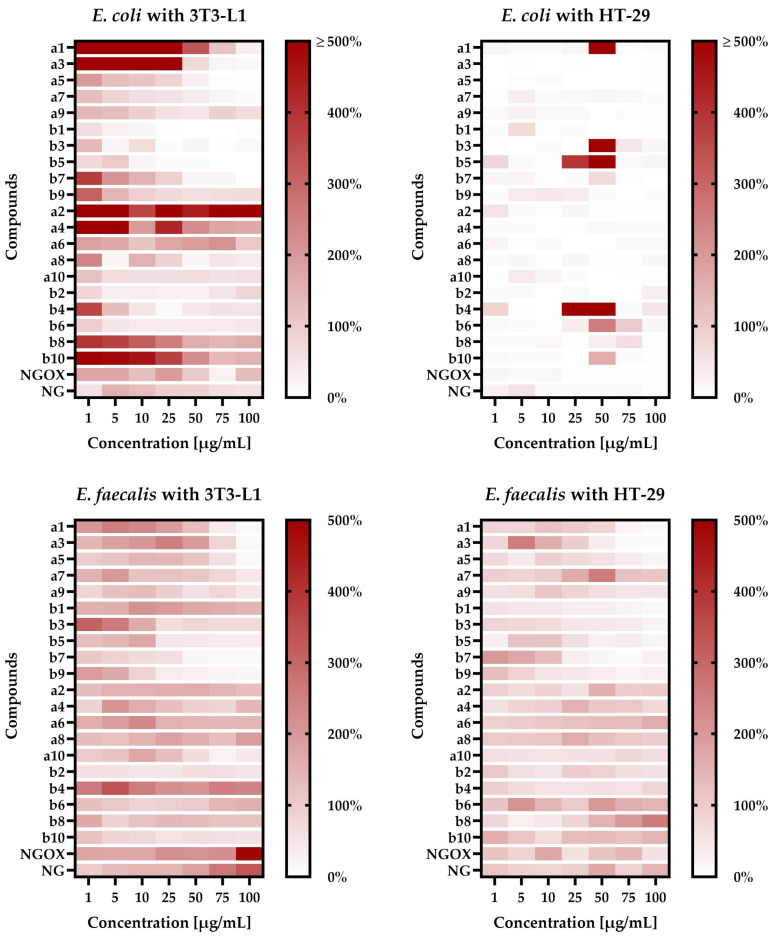
The results presenting the influence of the tested *O*-alkyl derivatives of naringenin (**1a**–**10a**) and their oximes (**1b**–**10b**) on the adhesion of *E. coli* and *E. faecalis* to the surface of the 3T3-L1 and HT-29 cell lines in static conditions. The data are presented in the form of heat maps. The study was performed with triplicate biological replications with three technical repetitions (*n* = 9); **NG**—naringenin; **NGOX**—naringenin oxime.

**Figure 2 antibiotics-12-01076-f002:**
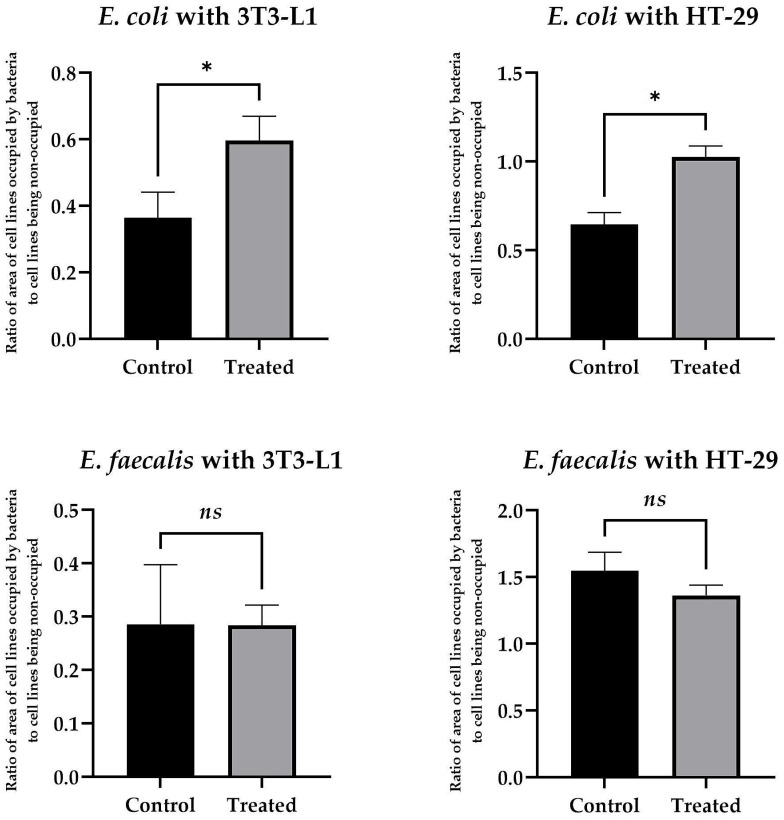
The results presenting the influence of **8b** at 50 µg/mL on the adhesion of *E. coli* and *E. faecalis* to the surface of the 3T3-L1 and HT-29 cell lines in the microfluidic conditions generated by the Bioflux system. The study was performed with triplicate biological replications with three technical repetitions (*n* = 9). The *p*-value represented by *ns* was statistically insignificant, while a *p*-value of <0.05 was considered statistically significant and presented as “*”.

**Figure 3 antibiotics-12-01076-f003:**
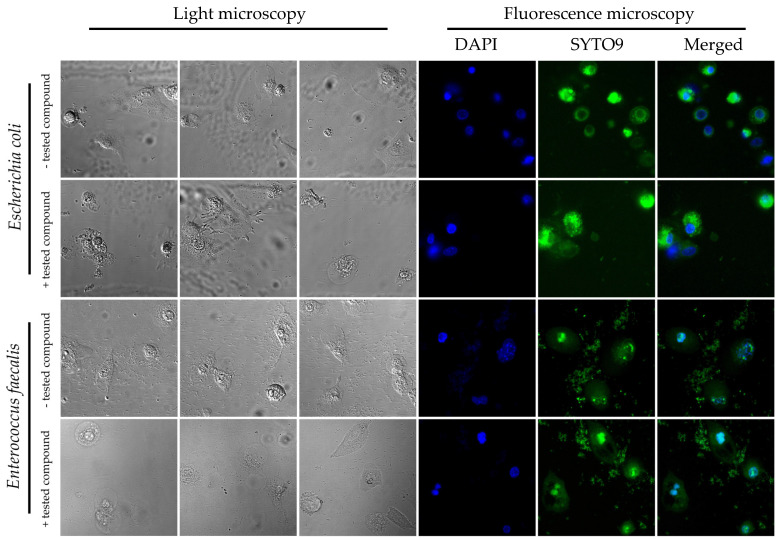
Representative photographs of light and fluorescence microscopy showing the adhesion of *E. coli* and *E. faecalis* to the surface of the physiological cell line, 3T3-L1, with and without exposure to **8b** (50 µg/mL), in the microfluidic conditions generated by the Bioflux system. DAPI stains the nucleus of the 3T3-L1 cell line, while SYTO9 stains bacterial cells (*E. coli* and *E. faecalis*).

**Table 1 antibiotics-12-01076-t001:** Structures of selected *O*-alkyl derivatives of naringenin (**1a**–**10a**) and their oximes (**1b**–**10b**).

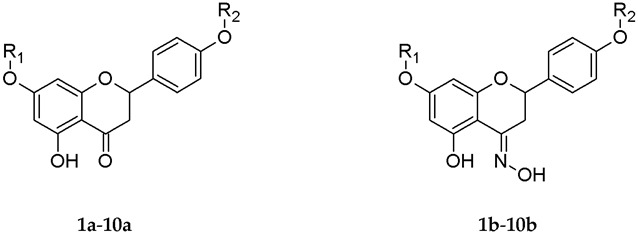		
**Compound**	**R_1_**	**R_2_**
**1a**-7-*O*-hexylnaringenin **1b**-7-*O*-hexylnaringenin oxime	hexyl	H
**2a**-7,4’-di-*O*-hexylnaringenin **2b**-7,4’-di-*O*-hexylnaringenin oxime	hexyl	hexyl
**3a**-7-*O*-heptylnaringenin **3b**-7-*O*-heptylnaringenin oxime	heptyl	H
**4a**-7,4’-di-*O*-heptylnaringenin **4b**-7,4’-di-*O*-heptylnaringenin oxime	heptyl	heptyl
**5a**-7-*O*-octylnaringenin **5b**-7-*O*-octylnaringenin oxime	octyl	H
**6a**-7,4’-di-*O*-octylnaringenin **6b**-7,4’-di-*O*-octylnaringenin oxime	octyl	octyl
**7a**-7-*O*-nonylnaringenin **7b**-7-*O*-nonylnaringenin oxime	nonyl	H
**8a**-7,4’-di-*O*-nonylnaringenin **8b**-7,4’-di-*O*-nonylnaringenin oxime	nonyl	nonyl
**9a**-7-*O*-undecylnaringenin **9b**-7-*O*-undecylnaringenin oxime	undecyl	H
**10a**-7,4’-di-*O*-undecylnaringenin **10b**-7,4’-di-*O*-undecylnaringenin oxime	undecyl	undecyl

Legend: H—hydrogen atom.

**Table 2 antibiotics-12-01076-t002:** Antiproliferative activity of *O*-alkyl derivatives of naringenin (**1a**–**10a**) and their oximes (**1b**–**10b**) against the HT-29 and 3T3-L1 cell lines, after 2 h incubation, tested with the SRB assay.

Compound	HT-29	3T3-L1	Compound	HT-29	3T3-L1
	IC_50_ [µg/mL]	IC_50_ [µg/mL]		IC_50_ [µg/mL]	IC_50_ [µg/mL]
**1a**	25.39	25.81	**1b**	38.44	34.99
**2a**	>100	>100	**2b**	>100	>100
**3a**	28.70	25.55	**3b**	29.15	32.12
**4a**	>100	>100	**4b**	43.35	>100
**5a**	38.35	20.09	**5b**	49.78	49.93
**6a**	>100	>100	**6b**	>100	>100
**7a**	46.72	22.23	**7b**	57.81	41.56
**8a**	>100	>100	**8b**	>100	>100
**9a**	>100	80.56	**9b**	>100	>100
**10a**	>100	>100	**10b**	48.23	>100
**NG**	>100	>100	**NGOX**	>100	>100

## Data Availability

All the data is included in the article.

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
