# Peer review of "Ether Derivatives of Naringenin and Their Oximes as Factors Modulating Bacterial Adhesion"

_antibiotics, 2023, doi:10.3390/antibiotics12061076_

Round 1

Reviewer 1 Report

I am happy to recommend the manuscript “Ether derivatives of naringenin and their oximes as factors modulating bacterial adhesion” by Anna Duda-Madej et al. for publication in Antibiotics. This is an interesting and well-written manuscript supported by scientific data, and the content of this manuscript is of high value to the antibiotics readers. In this work, the authors designed and synthesized O-alkyl Naringenin and their oxime analogues and evaluated the modulatory effect on E. coli and E. faecalis cell adhesion. Cell adhesion is the initial step in biofilm formation, and these naringenin analogues showed an important correlation between the O-alkyl chain length and concentration on cell adhesion. I would like to know the author’s future direction for this work.

Minor suggestions:

  1. Please provide the spectroscopic data (1H NMR and purity by HPLC) of the compounds tested.
  2. Regarding oximes analogues of naringenin, please comment on which isomer (E/Z) was obtained as the major product.   
  3. Reference formatting: Please write genus and species names in italics in the reference titles. Reference 53 citation, and DOI is also missing; correct reference 54.  

Author Response

We thank the Reviewer for the minor comments. We are very happy that you liked our Manuscript. In response to the Reviewer's comments, we would like to inform you:

1. Please provide the spectroscopic data (1H NMR and purity by HPLC) of the compounds tested.

Thank you very much for your comment. Detailed spectroscopic data (1H, 13C, 1H-1H (COSY) and 1H-13C (HSQC) NMR spectra and HRMS ESI-MS spectra) of all compounds were published in our previous work in the main text and also in the supplementary materials (doi.org/10.3390/ijms24129856). The purity of compounds was analyzed using NMR and HRMS ESI-MS techniques.

2. Regarding oximes analogues of naringenin, please comment on which isomer (E/Z) was obtained as the major product.   

Thank you very much for your comment. The formation of the Z-isomer was not observed because the E-isomer is thermodynamically more stable. The chemical part was described in detail in our previous Manuscript: doi.org/10.3390/ijms24129856

3. Reference formatting: Please write genus and species names in italics in the reference titles. Reference 53 citation, and DOI is also missing; correct reference 54

Thank you for the vigilance of the Reviewer. The DOI has been added to entry 53. In addition, the strain species names in the publication titles have been changed to italics.

We thank the Reviewer for his time and wish good luck.

Reviewer 2 Report

Some editions tahta can improve the quality of the manuscript: 

Table 1: abbreviations of O-alkyl and Oxymes substituents are confused. In the table have sufficient spaces to put or the molecular formula or complete name as Hexyl, octyl , etc. as other article  (doi:10.1016/J.EJMECH.2009.11.045)

Figure1: need to included the meaning of NGOx and NG.

Figure 2: the footnote it is not needed, it is the name of the compounds reflected in table 1. 

Conclusion: there are not new synthetized, there are published previously by the group. 

In vivo is in italic

Author Response

We thank the Reviewer for the time. In response to your advice, we would like to inform you: 

Table 1: abbreviations of O-alkyl and Oxymes substituents are confused. In the table have sufficient spaces to put or the molecular formula or complete name as Hexyl, octyl , etc. as other article  (doi:10.1016/J.EJMECH.2009.11.045)

Thank you for your advice. It was our oversight. We added the full names of the tested compounds in Table 1, while removing them from Table 2. Moreover, the names of the added residues were introduced in Table 1 based on the publication proposed by the Reviewer (doi:10.1016/J.EJMECH.2009.11.045). Thank you, we hope that now the Table 1 is more readable.

Figure1: need to included the meaning of NGOx and NG.

We thank the Reviewer for pointing out this inaccuracy. An explanation of the abbreviations used has been added.

Figure 2: the footnote it is not needed, it is the name of the compounds reflected in table 1. 

It has been moved to Table 1.

Conclusion: there are not new synthetized, there are published previously by the group. 

We thank the Reviewer for attention. The word "new" has been removed.

In vivo is in italic

Thank you for comment. We have followed the MDPI requirements (https://www.mdpi.com/authors/layout). We leave the final decision to the editors.

Good luck.

Reviewer 3 Report

In their work, the authors analyzed how the new synthesized ether and oxime naringenin derivatives affect and modulate representative samples of microbiota. These samples were E. coli and E. faecalis and they were tested on ‘healthy’ (physiological) and cancerous cell lines. The obtained results indicated that modulation of the adhesion depended on the concentration of the used naringenin derivatives and their structure (length of carbon chain in particular).

 This study is important because it could be a starting point for in vivo animal studies how these compounds affect the gut and what could be bioavailability and impact of these naringenin derivatives on the intestinal microbiome.  

 In this study the experiments were well planned and executed, the methods were well described even though, in Materials and methods section are necessary corrections: every chemical or instrument used needs to have a source (company, location, country), that has been done in most cases but not in all, it needs to be uniform. Also, this section needs minor English corrections.

Otherwise, the manuscript is well written and easy to read. Tables and figures are also well presented.

 The conclusions are appropriate and based on the obtained data presented.

References used in the manuscript are relevant and sufficient. 

 In this study the experiments were well planned and executed, the methods were well described, even though, in Materials and methods section are necessary corrections: every chemical or instrument used needs to have a source (company, location, country), that has been done in most cases but not in all, it needs to be uniform. Also, this section needs minor English corrections.

For example:

392 - Reaction mixture was kept on magnetic stirrer in room temperature (when reaction was performed in DMF) 393 or in 45 °C

at room temperature

at 45 °C

and similar.

Author Response

We thank the Reviewer for his time and very kind words. We improved our English in places recommended by the Reviewer, as well as in similar places. The prepositions have been changed to the appropriate ones. Missing information on the origin of the reagents used in the experiments has also been added.

Good luck
